# When animals cry: The effect of adding tears to animal expressions on human judgment

Alfonso Picó[1], Marien Gadea[1,2]*

**1** Department of Psychobiology, Faculty of Psychology, Universitat de València, Valencia, Spain, **2** Center of Network Biomedical Investigation - Mental Health (CIBERSAM), Madrid, Spain

* marien.gadea@uv.es

**Data Availability Statement:** All relevant data are available to view and download using the following link: https://osf.io/s9yev/?view_only=f91cccb72b5f4369bd7bf185764686c5.

## Abstract

At a time of growing interest in and awareness about the relationships between humans and animals, it is of relevance to scientifically analyse the intrinsic nature of these interactions. Reactions to emotional tears show our extraordinary capacity for detecting micro-nuances when judging another human's face. Regarding such behaviour, previous studies carried out in our laboratory have pointed to an adaptive function of emotional tears: i.e. their inhibitory influence on perceived aggressiveness. In the present work we aimed to further explore that hypothesis by extending our investigation from humans to animals, using pictures of five different animal faces (cat, dog, horse, chimpanzee, hamster) to which tears were added digitally. To this end, we conducted an online study of 403 participants recruited from different social networks and academic institutions. We questioned the participants about their perceptions of emotional intensity, aggressiveness and friendliness in the animal faces and analysed the comparisons they made between faces with and without tears. In addition, a latent variable referred to as "passion for animals" was measured using different indicators. By adding the results obtained in each species and breaking them down into different basic emotions, we found that the presence of tears was related to a higher absolute frequency of participants who perceived sadness, which endorsed our previous results obtained using images of humans. Regarding aggressiveness, the presence of tears favoured the perception of less aggressiveness. A structural equation model was also conducted to explore the relations among all the measured variables. The model confirmed that the presence of tears in the animal faces had a significant influence on the perception of higher emotional intensity and friendliness, and of lower aggressiveness.

## Introduction

Among primates, the detection of visual signals is crucial to recognize the emotions of others, being the eyes and the gaze the best sources for obtaining such an information [1–3]. Our large sclerae, accompanied by considerable eye mobility, make it easier to recognize the direction of the gaze and the object of attention of other humans, even in infants [4–6]. Such facilitated communication has several advantages for our theory of mind; that is to say, the

**Funding:** The author(s) received no specific funding for this work.

**Competing interests:** The authors have declared that no competing interests exist.

awareness of what another person is thinking or knows in contrast to our own experience, an essential skill in an ultrasocial species such as ours [7, 8].

The most particular and unique feature of our ocular physiology, and one that allows us to transmit a wide range of information in the form of emotional inference, is the shedding of emotional tears.

There is much scientific evidence to show that humans are extraordinarily apt at detecting micro-nuances that alter their emotional perception when judging another human's face, and our reaction to emotional tears is the perfect example [9]. Tears running down a cheek are capable of changing our bioelectric brain activity [10], and even alter the typical visual inspection pattern in such a way that tears act as magnets of attention [11].

While research on children's crying has focused on its acoustic component as a distress call [12], visual stimuli are the most widely used when studying its influence as a signal in adulthood. Interestingly, this distress call maintains its value as a sign of need for help throughout life, helping the crier to obtain the support of the group [13]. As a consequence of the perception of helplessness, people see a crier as more friendly, and are more willing to help them by increasing social connectedness [14]. In the present work, we explore another peculiarity of weeping with strong evolutionary roots.

Hasson hypothesized that tears serve as natural brakes to stop aggression in our conspecifics [15], and evidence obtained since then supports this theory [16]. The mechanisms through which visible tears achieve their appeasing effect on a potential aggressor are unclear, but they appear to be related to at least two different types of inference. On the one hand, (1) they have a significant influence on the perception of the crying person's vulnerability. The crier is not only perceived as more in need of help [10], thus awakening greater sympathy and empathy [17], but also generates a greater perception of incompetence [18]. On the other hand, (2) appeasement signals are only reliable if they prevent offensive actions in some way; however, by blurring vision, tears are a handicap in the case of having to defend against an attacker [15, 19]. Such appeasement signals are more likely in situations of hierarchical restructuring and other types of intra-group social contexts among equals, in which mortal aggression is unusual, where showing submission is an effective mean of communicating the cessation of agonistic behaviour.

A clever form of manipulation in order to study the function of tears and crying is the use of "poker faces"–faces which show little or no emotion at all [20] while crying (shedding tears); in this way, researchers can observe the effect of tears isolated from the emotional inference that usually accompanies marked emotional expressions. Among these studies, of which there are few, some have used calm crying expressions (see [21] for an in-depth definition) or human-like avatars [22, 23]. In this sense, and regarding the objectives of the present work, we believe that a new experimental way to further explore the effect of emotional tears on the receiver is to add visible tears to the face of different species of mammals, after first providing images in which the animals have a regular expression typical of its species. Animals represent an excellent opportunity to study the effect of tears on human observers, because they do not cry in response to emotional feelings (as commented above, emotional crying is a human feature; see [24] for an evolutionary overview). In this context, it is important to point out that the animal species selected for this study are all physically capable of shedding tears, but do not use this mechanism as an emotional signal to conspecifics [25]. In this way, such emotional signalling, if observed through human judgement, would represent a process of anthropomorphization.

The concept of anthropomorphism refers to the attribution of human-like properties, characteristics, or mental states to real or imagined non-human agents and objects [26]. It is a core concept discussed in HRI (human robot interaction) literature and, in general, in studies

describing how humans relate to their computers and other media technologies (see the seminal proposal of the Media Equation theory, which proposes that humans interact with their technological devices "as if they were humans too" [27]). People tend to anthropomorphize robots more than other technology, although this process varies depending on factors such a movement and gestures, verbal communication, embodiment and, most importantly for our interests, the emotional response shown by the robot [28]. In fact, a robot is perceived as more anthropomorphic when it provides emotional feedback, rather than when its feedback is unemotional [29]. Regarding the psychological process of anthropomorphization itself, several comprehensive theoretical accounts have been suggested, especially in the abovementioned HRI context. Among these, a "dual model" has been proposed, in which an implicit, automatic and unconscious process of attributing human-like qualities to non-human agents could coexist with an explicit, cognitive-driven, motivated and conscious process in the same direction. An important implication of this is that the direct measures of anthropomorphism (e.g. questionnaires) can reflect either implicit or explicit anthropomorphism, or a mixture of both, depending on the person [26].

A process of anthropomorphization in humans when describing animals is known to exist, and has a longer history, with Charles Darwin providing an excellent illustration of the human mind's capacity to see the same kinds of covert emotional states in the behaviour of non-human animals as it does in the behaviour of other humans [30]. This is unlikely to have an exact correspondence with the phenomenon of interaction with robots, given that animals have their emotions in physiological terms (but no feelings, according to their lack of a complex conscious behaviour). However, when we perceive that an animal is experiencing a concrete emotion or feeling we attribute a human feature to the animal, in which case we are indeed applying our anthropomorphism to the animal. Interestingly, this process can provide advantages to the animal, since individual differences in anthropomorphism predict the degree of moral care and concern afforded to an agent [31] and, moreover, empathy towards animals and humans is correlated [27]. In addition, our daily experience interacting with different types of animals has a long history and is much more extensive than the sporadic interactions we have with modern day avatars that express different emotional states, or non-generalized (till now) relationships with robots and other AI devices. Moreover, in line with this interest in the anthropomorphization of animals, some authors have explored its neural basis to show that we use the same neural mechanisms to attribute emotions to the facial expressions of humans and non-human animals [32].

In summary, by exploring how digitally added tears can change our judgments and inferences about animal faces, we have sought to take a step further after our previous work in which we added tears to human faces by opening up new lines of research into the psychological process of anthropomorphization, and increasing understanding of our perception towards other species' emotions, which partly conditions human-animal interactions. The measurement of some human-like attributions (human emotions or traits, like "sadness" or "aggressiveness") in regular animal faces can provide information about the process of anthropomorphization towards animals, while the addition of digital tears to their faces (making animals prone to be perceived as if they were crying) can enlighten us about the power of the presence of tears for such a process.

## Objectives and hypothesis

In light of the abovementioned literature, we aimed to further explore the putative adaptive function of an inhibitory influence of tears on perceived aggressiveness by using pictures of animal faces with digitally added tears. In general, we expected the perception of emotional

intensity in the facial expression to be greater when judging the "crying animals". More specifically, we hypothesized that adding tears to the face of an animal would arouse that the animal would be perceived as more human-friendly. As a consequence, we expected that, when participants judged aggressiveness, crying animals would obtain a lower score. In addition, we explored whether different species of animals inspired different impressions when seen to be weeping. For example, we speculated that the species which are most popular among humans, such as domestic pets (i.e. dog, cat, and hamster), would have a differential impact on observers when their faces had visible tears. Finally, since we aimed to determine the most relevant variables related to crying, we decided to evaluate our data through a structural equation model that contemplated how previous fondness for animals related to the variables assessed.

## Materials and methods

### Participants

Four hundred and three participants aged between eighteen and sixty-four years old—94 males (M = 33.95, SD = 10.52) and 309 females (M = 31.61, SD = 11.04)—took part in the study as volunteers, without receiving any reward. To test our hypotheses, we conducted an online study in which participants were recruited from different social networks and academic institutions via different methods (e.g. newspaper ads, social networks and announcements during university classes). Participants were given a link to a survey administered via Google forms and the random redirector allocate.monster. Subjects were treated in accordance with our university's Ethical Code of Conduct. Informed consent was obtained from all the participants, and all procedures were in accordance with the standards of the *Comité de Ética de Investigación en Humanos* (CEIH) from the University of Valencia (Spain), which approved the study, and with the 1964 Helsinki declaration and its later amendments.

### Visual stimuli

The stimuli consisted of a set of photographs depicting close-ups of the faces of five different animals: a chimpanzee, a horse, a dog, a cat, and a hamster. The pictures were obtained from Google and were free of copyright. These original images were then modified by digitally adding visible tears with Adobe Photoshop Cc 2019. This yielded two groups of five pictures that were identical except for the presence (or not) of tears. An example of the same photograph (dog) with and without tears can be seen in Fig 1. The rest of the photographs can be found in the supplementary material. In addition, in order to evaluate the quality and characteristics of the stimuli employed, an experimental study was carried out (described in detail in the supplementary material; see link below). The added tears were shown to be easily visible, and all the selected animals were perceived to be typically representative of their species. Interestingly, the original image of the dog was perceived as emotional, hinting that, at least in the case of man's best friend, a certain emotional inference can arise without the presence of tears. These latter results, together with the results of the main study, will be discussed in the context of their relevance to anthropomorphization and emotional inference tasks.

All supplementary materials can be freely consulted in https://osf.io/s9yev/?view_only=f91cccb72b5f4369bd7bf185764686c5.

### Measures

After being presented with both versions of each photograph (with and without tears) the participant responded to a questionnaire regarding his/her impressions: (1) what type of emotion (if any) was expressed by the animal; (2) the intensity with which the emotion was expressed;

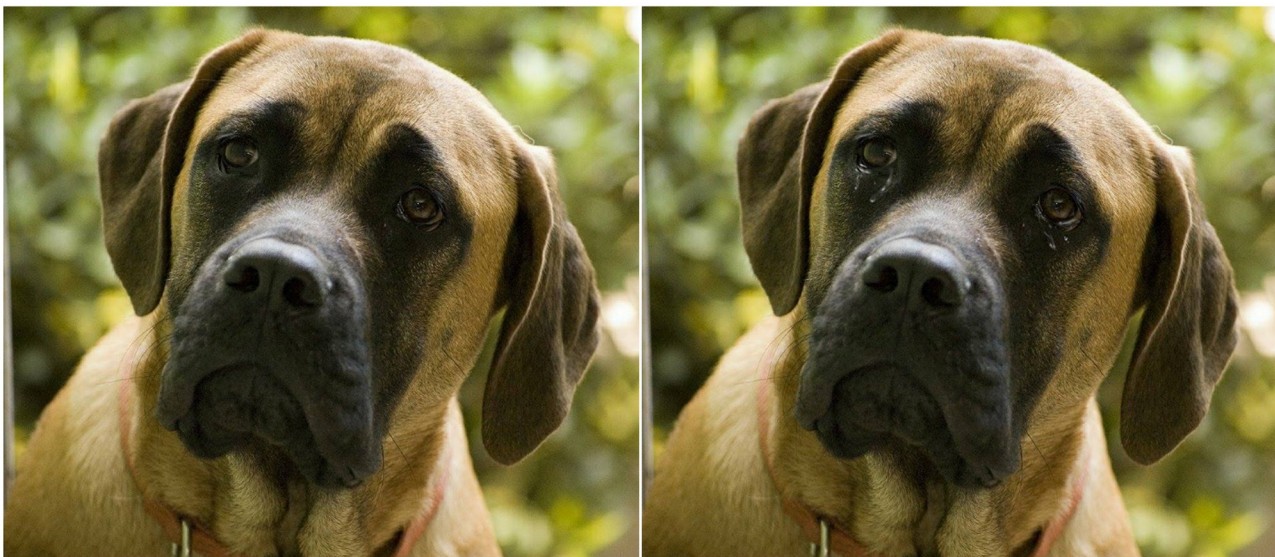

**Fig 1. The original photograph of a dog's face and its tearful version.**

(3) the friendliness perceived from the animal (considering "friendliness" to mean the tendency to perform acts of kindness towards others); and (4) the aggressiveness perceived from the animal's expression.

To assess the perceived emotional expression (question 1), participants were asked to select a basic emotion (i.e. anger; surprise; happiness; fear; disgust; sadness) if they thought the animal was expressing such, or to indicate that the animal's expression was emotionally "neutral" if they felt that was the case. We then applied a dichotomous classification based on previous works: emotion, if any, versus emotionless [11, 16]. "Emotionless" indicated faces with no perceived expression. To assess the remaining variables (questions 2-3-4), they were scored according to a 10-point scale, where 0 indicated the complete absence of intensity, friendliness or aggressiveness, and 10 the highest degree of these emotions. In addition, prior to the experimental procedure, participants were asked to answer 3 questions concerning their relationship with animals. (1) How much do you like animals? (on a scale of 0 /I don't like them at all to 10/ I am really fond of them; (2) How many animals have you lived with? (a scale of five, from none to one, two, three, or more than three); (3) How much importance have animals had in your life until now? (from 0/absolutely nothing to 10/extremely important). These three questions were used as indicators of a latent variable which we labelled "passion for animals".

## Procedure

After being recruited and obtaining their informed consent, participants answered the three questions about their relationship with animals. Participants were randomly assigned to one of two groups depending on the presence of tears in each animal model. In this way, Group 1 was presented with the tearful chimpanzee, the tearless horse, the tearful dog, the tearless cat, and finally the tearful hamster. For group 2, the presence/absence of tears was inverted. Thus, although all participants were presented with animal faces with and without tears, each participant saw only one version of each picture, which resulted in a between-subjects experimental design. In addition, the order of the pictures was randomly assigned for each participant; subsequently, he/she was presented with a picture of the first of the animal models and then

completed an online questionnaire about its face, which was answered without a time limit. This process was repeated four times until the participants had been presented with the five animal pictures. A visual presentation of the procedure can be seen in Fig 2.

### Statistical data analysis

To assess perceived emotionality, we first used a dichotomic discretization: emotional (if any emotion was detected) versus emotionless. Emotionless indicated faces with no perceived expression at all. Fisher's exact test for contingency tables [2 (tearless or tearful) x 2 (emotional or emotionless)] was used to calculate the odds ratio of perceived emotion on a face regardless of the species of animal. We also disclosed the odds ratio separately by species. Next, a 6 x 6 matrix of Pearson's correlation coefficients was used to evaluate separately the strength and direction of the association between the variables measured (intensity of emotion, friendliness, aggressiveness) in the set of original photographs depicting the animals without tears and in the set of photographs with digitally added tears (the five species were pooled for this evaluation). We then carried out separate ANOVAs for each species and calculated post-hoc comparisons when required. Each of the former statistical analyses was carried out with R v.3.6.1 software. Finally, we performed and fitted a structural equation model to explain the causal relationship among variables by applying Mplus version 8 and a maximum likelihood sandwich estimator with robust standard errors [33].

## Results

### On emotionality and tears in animals

Considering the effect of visible tears on human faces, we expected that adding tears would dramatically raise the perceived emotionality of our models' faces. First, we used the pooled species data and found that they were in fact associated with emotionality (odds ratio = 1.41, $p$ < .001). Then they were detailed when analysing the effects of tears by species, to find that chimpanzee (odds ratio = 2.15, $p$ < .01), horse (odds ratio = 2.60, $p$ < .001), and dog models (odds ratio = 2.28, $p$ < .01) reached statistical significance when judged as more emotional in the tearful condition. The hamster did not show significant differences (odds ratio = 1.14, $p$ = ns), while the opposite was true for the cat (odds ratio = 0.55, $p$ < .01).

In order to visually explore the above mentioned results to check if such a generally increased perception of higher emotionality was due to any particular basic emotion, we depicted the Fig 3, to find that the presence of tears was related to a higher absolute frequency of participants who perceived sadness (see Fig 3). Moreover, the absence of visible tears was frequently associated with a neutral expression and, interestingly, with an angry emotion (although later visual inspection of the data showed this latter observation was mainly due to the tearless image of the cat).

### On intensity of emotions

All the animals, except the cat, were associate with a higher mean intensity of emotions when tears were visible (Fig 4). In the dog's case [$F_{(1, 400)}$ = 16.41, $p$ < .001] a significant difference was found in the post-hoc test, with a Tukey HSD of 1.23. In the same way, the chimpanzee [$F_{(1, 400)}$ = 5.80, $p$ = .016] was perceived to be much more emotional when tears were visible (Tukey HSD of 0.65). Regarding gender differences, women rated the cat and the horse models with greater emotional intensity than men [$F_{(1, 400)}$ = 10.10, $p$ = .001] and [$F_{(1, 399)}$ = 6.61, $p$ = .010] (*Tukey HSD* of 1.29 and 0.5, respectively). Interestingly, in the case of the horse [$F_{(1, 399)}$ = 5.12, $p$ = .024], there was a significant interaction between the presence of tears and the

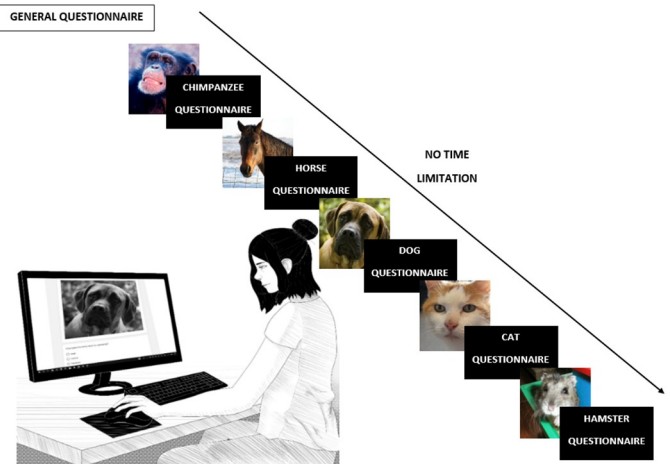

**Fig 2. Visual outline of the procedure.** Participants answered a general questionnaire about their relationship with animals and were then presented with five photographs of the faces of five different animals. While each photograph was on the screen, participants answered four questions about the face in question, without any time limit being imposed. Participants had been randomly assigned to two different groups depending on the presence of tears in the photographs. Group 1 saw the tearful chimpanzee, the tearless horse, the tearful dog, the tearless cat, and finally the tearful hamster. For group 2 the presence/absence of tears was inverted, so that they saw the tearless chimpanzee, the tearful horse, the tearless dog, the tearful cat, and the tearless hamster.

participant's gender. In the tearful condition, women found the horse to be much more intense than men, while men rated the picture with more intensity than women when tears were absent. Although both genders were sensitive to the tearful effect, the trend was more pronounced among women.

## On aggressiveness

Our hypothesis about less perceived aggressiveness is supported by the most consistent result of our experiments. The presence of tears was a significant influence in four of the five animal

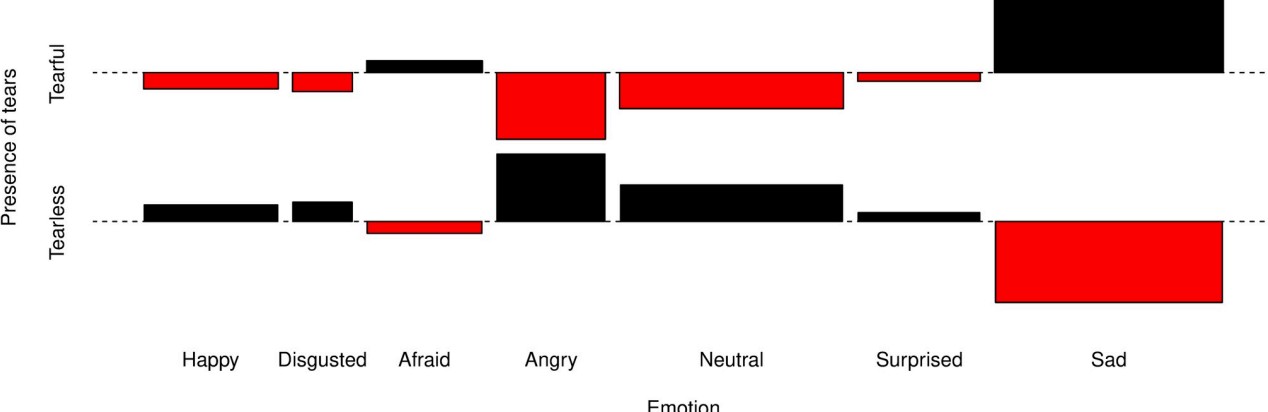

**Fig 3. Cohen-friendly association plot.** For a two-way contingency table, the signed contribution to Pearson's $\chi^2$ for the cell $ij$ is $d_{ij} = f_{ij} - e_{ij}/\sqrt{e_{ij}}$, where $f_{ij}$ and $e_{ij}$ are the observed and expected counts corresponding to the cell. In this association plot, each cell is represented by a rectangle whose height is proportional to $d_{ij}$ and whose width is proportional to $\sqrt{e_{ij}}$, so we can see that the area of the box is proportional to the difference in observed and expected frequencies [34].

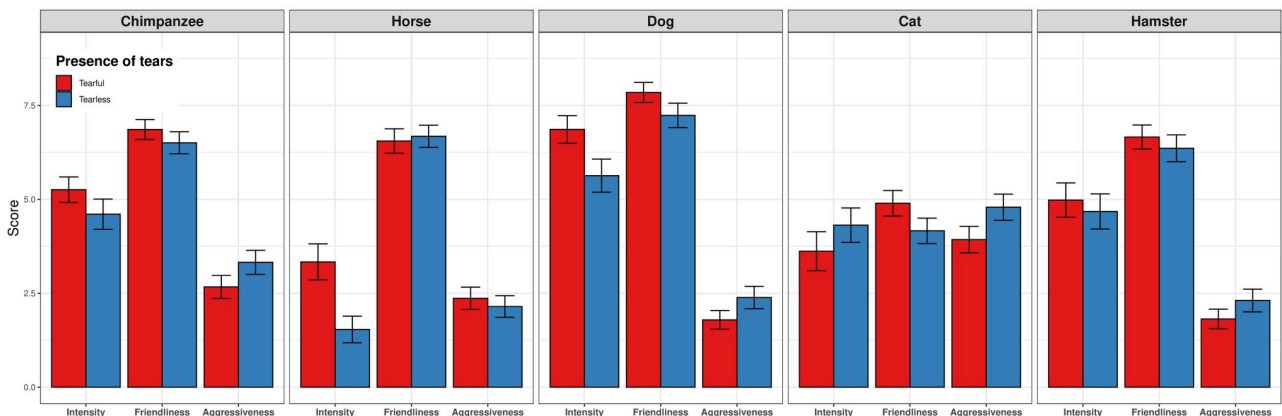

**Fig 4. Bar plots per species and presence of tears.**

species when humans judged their faces (Fig 4); namely, the chimpanzee [$F$ (1, 400) = 7.58, $p$ = .006], dog [$F$ (1, 400) = 7.94, $p$ = .005], cat [$F$ (1, 400) = 12.18, $p$ < .001] and hamster [$F$ (1, 400) = 5.08, $p$ = .02], with Tukey's honestly significant differences of between 0.49 and 0.85. In addition, there were main effects of the participants' sex on the horse and dog models. Women rated aggressiveness lower than men, with $F$ (1, 400) = 5.52, p < .05 (*Tukey HSD* of -0.56) and $F$ (1, 400) = 4.50, p < .05 (*Tukey HSD* of -0.48), respectively.

## On friendliness

All animals except the horse showed a higher mean friendliness when tears were visible (Fig 4), though only two reached statistical significance: the dog [$F$ (1, 400) = 6.99, $p$ = .008] and cat [$F$ (1, 400) = 10.01, $p$ = .001]. Post-hoc tests revealed a significant difference in favour of the tearful condition for both animals(crying animals were perceived as more friendly), with a Tukey HSD of between 0.65 and 0.73. Gender also proved to be a significant factor when the horse [$F$ (1, 400) = 6.21, p = .01] and dog [$F$ (1, 400) = 5.22, p = .023] were rated, with a *Tukey HSD* of between 0.59 and 0.65. Women perceived the tearful pictures of both the horse and dog to be more friendly than did men.

A summary of the means and their standard errors for the five species studied, with and without tears, and the three main dependent variables (intensity of emotion, perceived friendliness, and perceived aggressiveness) is provided in Fig 4.

## Correlations between variables

As shown in Table 1, there was a strong correlation between love for animals and the importance of animals in our participants' lives ($r$ = .726, $p$ < .001), and also between the importance of animals and the number of animals a participant had lived with ($r$ = .493, $p$ < .001). In addition, number of animals and love for them were significantly correlated ($r$ = .422, $p$ < .001), which was expected. Another interesting correlation emerged between the intensity of the emotion perceived and friendliness ($r$ = .222, $p$ < .001), while a significant association was not detected between intensity and judged aggressiveness ($r$ = -.021, $p$ = ns). In this context, one would expect a significant negative correlation between friendliness and aggressiveness ($r$ = -.380, $p$ < .001).

**Table 1. Matrix of Pearson correlations with the means of the five species.**

|  | Number | Love | Importance | Intensity | Friendliness | Aggressiveness |
|---|---|---|---|---|---|---|
| Number | - |  |  |  |  |  |
| Love | **.422***** | - |  |  |  |  |
| Importance | **.493***** | **.726***** | - |  |  |  |
| Intensity | -.039 | **.080**** | **.087**** | - |  |  |
| Friendliness | **.068**** | **.156***** | **.135***** | **.222***** | - |  |
| Aggressiveness | -.020 | **-.074**** | **-.084**** | -.021 | **-.380*** | - |

Note.

* $p < .05$,

** $p < .01$,

*** $p < .001$.

All p-values were corrected using the Holm-Bonferroni method for multiple comparisons.

### Structural equation modelling of perception of aggressiveness

As demonstrated in Fig 5, indicators showed that the model fit the data in a satisfactory way, with a RMSEA of 0.04 (i.e. a close fit according to the guides [35–37], a CFI of 0.988 (well above the recommended value of 0.90) and a SRMR of 0.02 (considering a SRMR < 0.08 to be a good fit), although the $\chi^2$ statistic obtained for the model was 20.39 ($df = 9$, $p = .015$), which suggested a non-perfect overall fit of the model. We also tested the model by adding the variable Gender, but the indices of fit in said model were substantially lower, so we decided to present the more parsimonious first model, which had obtained the closest fit. The trend towards perceiving animals to be more friendly had an influence on the passion for animals, but the most stand-out finding was that tears had both direct and indirect effects on the perception of aggressiveness.

Note that a table of means and standard deviations (or its bar plot version) per species, experimental condition (tearful vs tearless), and gender can be found as supplementary material at the link provided above (see Visual Stimuli), along with the visual stimuli, mixed effects models (as an alternative to the classical analysis presented), and the database.

## Discussion

The main aim of this study was to assess the influence of visible tears on the faces of non-human animals on our perception of aggressiveness, friendliness, and emotional intensity.

As mentioned in the introduction, the shedding of emotional tears is a solely human feature, as non-human animals do not generate tears to express emotions, though they do display their emotional state in other ways [1, 38]. Our first analysis, in which we asked participants if "the animal is showing any emotion" confirmed the initial hypothesis: namely, tearful animals are perceived to express more emotionality. Thus, it was not necessary for the animals to shed emotional tears for human participants to perceive them as emotional, especially in the case of the dog, chimpanzee and horse models. Interestingly, our survey about the quality of the tearless images rendered a median rating of around 5 for every species except the dog, in a 0–10 points scale where 0 was "emotionless" and 10 "absolutely emotional". Therefore, as expected, adding tears significantly increased the probability of human observers engaging in an anthropomorphization process towards the animals, attributing them an emotional expression they did not really have. The case of the dog was unique, as it was rated emotional even without tears, while the presence of tears increased the attribution of emotionality (this will be

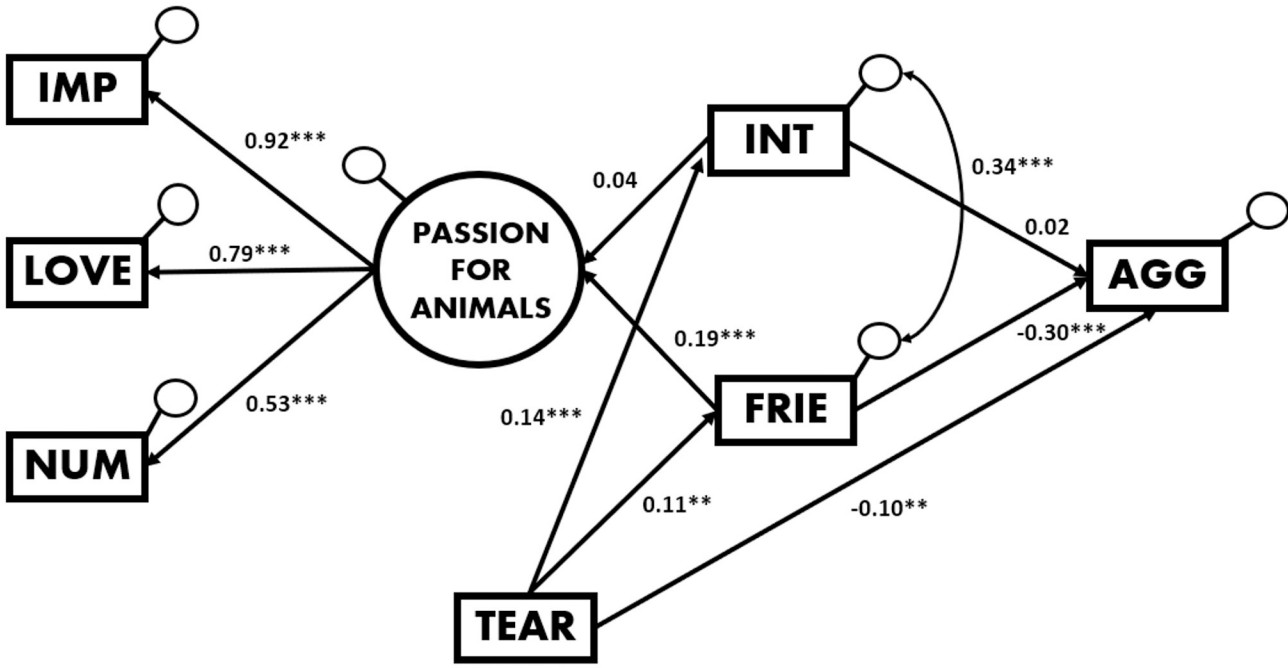

**Fig 5. SEM of perception of aggressiveness with mediational effects.** Standardized estimates with two-tailed p-values. The importance that animals have in our participant´s life (IMP), the love felt for them, and the number of animals with which participants have lived are indicators of the latent variable labelled "passion for animals". This passion is significantly and positively influenced by the trend towards perceiving animals to be friendlier, while the presence of tears is a significant influence on the intensity, friendliness, and aggressiveness perceived. As illustrated, tears not only have a direct effect on the perception of aggressiveness, but also indirectly, through the friendliness perceived. CFI = 0.988, RMSEA = 0.04, SRMR = 0.02, $\chi^2(9)$ = 20.39 with $p < .05$. $^*\ p < .05$, $^{**}\ p < .01$, $^{***}\ p < .001$.

discussed below, with regards to domestic animals and their particularities). In terms of the type of basic emotion assigned to the tearful animals, sadness was the most frequently chosen. The fact that the presence of tears is related to an increased frequency of perceived sadness supports previous results obtained using images of humans [10, 21]. In our second analysis, when participants were asked about the intensity of the emotion perceived in the animals' faces, we observed that the tearful animals were judged to be more intensely emotional, which is again in line with the above mentioned results. An exception was made for the cat, which will be discussed below with regards to domestic animals.

The results of our analyses of friendliness and aggressiveness extend our previous work on tears as appeasement signals [16] by highlighting a clear and widespread trend towards judging tearful animal models to be less aggressive. As we expected, the emotional stimulus (fake tear-drops descending from the eyes to the cheeks of the animals) exerted a significant influence on the perception of the observers, and almost every animal model was perceived to be less aggressive when such tears were added to its face. Given that the study about the quality of the images indicated that the tears were perceived as easily visible and mostly realistic, we can argue that the observers judged the animals to be crying naturally, and displayed an emotional reaction to them in the line of what the literature reflects about the effects of tears on humans and their adaptive functions [14]. In addition, the results of our structural equation modelling analysis (SEM) supported our hypothesis: that there is a negative correlation between judging an animal to be more friendly and perceiving it to be less aggressive. In other words, we expected the two perceptions to be antagonists [39]. Indeed, friendliness and aggressiveness are associated with different physical gestures and involve distinct neural circuits [40]. This

negative correlation was statistically significant in our study, and the structural equation modelling—which displays the pure effect of friendliness on aggressiveness—confirmed our expectations. Thus, in our causal model it was shown that tears influenced perceived aggressiveness in a direct way, causing the animals with tears to be perceived as less aggressive, and in an indirect way, through the expression of friendliness. Another interesting finding of the SEM analysis is the absence of a significant influence of emotional intensity on the perception of aggression. In fact, the presence of tears was a significant influence on perceived emotional intensity, but the latter did not affect judgment of aggression. Thus, we can affirm that, based on our structural equation model, emotional intensity does not affect perceived aggressiveness in any way, although emotionality is perceived in facial expressions more frequently when tears are present. Ito, Ong, and Kitada [41] found that the mere presence of tears on human faces with neutral expressions makes the viewer more prone to perceiving sadness, a result that we extend to different animal species in our study. In addition, our SEM analysis indicated that the latent variable labelled "passion for animals", a compound of the relative importance people give to animals, the love they show them, and the number of animals they own, is positively influenced by the friendliness that people perceive in animal faces. In summary, and considering the results as a whole, we can affirm that human observers transfer some of the purported appeasement functions of the non-verbal communication signal of tearing to other species.

Regarding the pictures of the cat and the dog, we think they deserve some words apart, given they are the most common domestic animals. In this sense, both transmitted the appeasement effect of tears and were judged to be less aggressive in the tearful pictures, but we also observed a few interesting particularities which distinguished them. The dog was seen as more intensely emotional and to be less aggressive than the cat, and this was observed even in the tearless picture. The cat was judged to be the most aggressive animal and, surprisingly, to have a less emotional expression (less intense) when depicted with tears. To provide with a speculative reason as to why this difference occurred requires a deep understanding of the particularities of our relationship with dogs and cats. Both species have shown themselves to be capable of flexible learning, which refers to the ability to adapt their repertoire of learned behaviours to circumstances; however, early studies of comparative psychology [42] showed dogs to be much easier to train than their feline counterparts. Naderi et al. [43] believed that this difference was due to the eye contact dogs make with their trainers, since humans have preference for this type of exchange (e.g. gaze contact), and dogs possess the natural advantage of using the same visual signals as humans, with evidence showing they are capable of processing facial emotion [44]. This could explain how experience has resulted in a preference among humans to make intense eye contact with dogs (which dogs return), while interaction with cats is not subject to the same contingencies. In relation to this, it has been proven that domestic cats make use of vocalization, exploiting certain human sensory biases in relation to the acoustic component of babies' crying to obtain attention and food [45]. Dogs also produce different vocalizations according to the emotional weight of the information they want to transmit [46], but visual communication seems to be favoured, especially in the case of some breeds [43, 47]. We suspect that this routine interaction with certain species that have coexisted closely with humans for thousands of years has sensitized us to a differential communication with animals. In other words, we might be more accustomed to observing the faces of dogs than those of cats or other species to try to infer their emotional state, and thus the process of anthropomorphization may arise more readily in our interaction with dogs (in this context, we feel it is not a coincidence that both the robots Spot, from Boston Dinamycs, and Miro, from Consequential Robotics, are dog-shaped). On the contrary, cat owners have problems identifying the emotional valence of their companions' faces in an emotional inference task [48]; at best, they can

interpret the emotional valence of their own cat's signals in a reliable fashion, but not those of unfamiliar cats [49]. In any case, the reader should note that this vision of domestic animals together with our interpretation is tentative in nature and calls for caution when applying to actual relations between humans and animals. In addition, the differences observed here could also be due to any remaining particularities of the chosen image (breed, fur, natural expression of the concrete animal in the photograph. . .).

With respect to a few gender differences observed, women perceived a higher emotional intensity in the case of the cat and horse. There is evidence of a preference for horses among girls, while boys prefer wild animals [50], and females are more likely to label themselves as cat persons [51]. Perhaps the gender effect in terms of perceived emotional intensity can be explained by a training effect, since women are more likely to spend time searching for videos of cats and horses on the internet, and will therefore tend to be more used to interpreting the emotional intensity of these animals [52]. In addition, our women not only perceived the dog and horse to be more friendly, but also found them less aggressive. Such an effect with respect to dogs could be moderated by hormones, as there is a different response in oxytocin levels between men and women when they interact with their dogs [53], but this theory requires more empirical support.

Finally, we noted a few unexpected results (the "colder tearful cat" and the absence of a tear effect on aggressiveness in the horse) that were obtained with the pictures that were rated as less realistic in the quality study. This prompts an interesting reflection on the use of artificial stimuli to depict human emotions in non-human targets (animals and possibly robots), in the sense that, the more artificial and unreal they are perceived to be, the more difficult it is for them to produce empathy in us, and thus an anthropomorphic feeling.

## Strengths and limitations

This research demonstrates the advantage of an adequate sample number that provides the desired power. Furthermore, as far as we know, this study is the only one in which animal images have been manipulated in order to explore the effect of visible tears on emotional and moral inference while isolating them from any explicit expression, thus allowing the influence of tears on human observers to be measured. The present findings are strengthened further by the fact that the animal faces were heterogeneous. However, a possible limitation of the present work is the small number of images used, given that a greater number of images of each species could have avoided the potential influence of the particularities of each photograph and species. The study on the quality of the images showed a few limitations like lack of perfect realism for some stimuli (horse, cat) and a variability in the extent to which some of the tearless stimuli were perceived as emotional (especially the dog). On the other hand, an excessive number of photographs could have affected the results, as tears can lead to a rapid habituation of the viewer; in this case the possibility of obtaining a non-significant result would increase. In addition, Prokop and Fancovicova [54] have found that animal species and their colours are important predictors of the response they provoke in human children, and it is possible that tears do not exert an effective influence in the faces of animals that evoke danger and disgust without tears.

## Future research

In this study, the presence of tears on animal faces altered the emotional inference and perception of aggressiveness in our human judges. Previous studies have shown that certain animal species are sensitive to visual information provided by human facial expressions [e.g. 44, 53, 55, 56]. A further step for future research would be to assess whether these animals also

respond to the minimal visual signal of emotional tears; specifically, future studies could present dogs with photographs of their owner's with and without emotional tears. Also, and in order to explore in further depth the emotional inference of animal faces for human participants, it would be interesting to test the effect of animal age on our perceptions. According to Murube [57], emotional tears are the last to appear in the phylogeny and ontogeny of our species (preceded by basal and reflex tears), and our moral judgments may mimic this phenomenon by giving greater social value to tears emitted in adulthood. Human babies do not emit tears in response to affective stimulation until they are at least 6 weeks old [58], and this behaviour does not fully develop until the efferent parasympathetic pathway and supranuclear nerve connections mature [59], which usually occurs at 4 months. The gestures and sounds of emotional crying are more important in the early stages of childhood [60], while the visual component gains stimulus salience as the individual enters adolescence and adulthood. In addition, as Zeifman and Brown [12] reported, the significance of the changes in moral inferences produced by tears increases as we enter adulthood. In this way, babies benefit less from the presence of tears than older children, and the latter benefit less than adults. It would be interesting to determine if age influences human emotional inference with respect to animal faces without tears, as it does when tears are added.

## Conclusions

Emotional tears are a special form of non-verbal communication that are unique to humans and which favour the inference of emotional states. They have adaptive advantages; among these, we have hypothesised that tears are an honest biological signal with a clear purpose of inhibiting aggression towards the crier in social contexts. In the present study we have extended the universality of this hypothesis by using animal faces to which we artificially added visible tears. By means of this experimental methodology, and a causal model for explaining the relationships among the variables evaluated, we provide empirical support for the notion that the presence of artificial tears on the face of an animal results in the human observer perceiving it to be less aggressive, possibly through a process of implicit anthropomorphization. In addition, such tears increase the perceived friendliness of the animal, and this influences the perception of its aggressiveness. Moreover, our results show that our passion for animals, a variable which includes how much an individual likes animals, how many animals he/she is living with, and how important animals are for the individual, is clearly affected by our perception of the friendliness of the animal, which increases with the presence of tears. Considering the results as a whole, it seems that the presence of tears improves the social relationship between humans and animals. Given that animals do not cry in natural conditions, the results observed here are presumably a consequence of some anthropomorphization process between the human observer and the animal, which leads us to propose that tears are the main element in this improved social communication. In conclusion, our results further endorse the notion of tears as an important biological signal that is essential for non-verbal communication.

## Acknowledgments

We want to thank the digital artist María Fernandez Peris, whose contribution was fundamental when modifying the animal images in order to add visible tears and during the creation of some of the figures. We also want to thank the PhD student Shinji de Paula for his collaboration in recruiting part of the sample.

## Author Contributions

**Conceptualization:** Alfonso Picó.

**Data curation:** Alfonso Picó.

**Methodology:** Alfonso Picó.

**Writing – original draft:** Alfonso Picó, Marien Gadea.

**Writing – review & editing:** Alfonso Picó, Marien Gadea.

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
