## [Decision Letter · Decision Letter 0]

9 Dec 2020

Pécs, Hungary

December 8, 2020

PONE-D-20-35135

When Animals Cry: The Effect of Adding Tears to Animal Expressions on Human Judgment

PLOS ONE

Dear Dr. Gadea,

Thank you for submitting your manuscript to PLOS ONE. After careful consideration, we feel that it has merit but does not fully meet PLOS ONE’s publication criteria as it currently stands. Therefore, we invite you to submit a revised version of the manuscript that addresses the points raised by the Reviewers, listed below.

We look forward to receiving your revised manuscript.

Kind regards,

Joseph Najbauer, Ph.D.

Academic Editor

PLOS ONE

Journal Requirements:

2. Thank you for including your ethics statement:  "The study was under the standards of the Human Research Ethics Committee of the University of Valencia (Spain) and a written informed consent was obtained from all the participants.".   

Please amend your current ethics statement to confirm that your named institutional review board or ethics committee specifically approved this study.

4. Please include a copy of Table 1 which you refer to in your text on page 12.

Reviewers' comments:

Reviewer's Responses to Questions

**Comments to the Author**

1. Is the manuscript technically sound, and do the data support the conclusions?

Reviewer #1: Partly

Reviewer #2: Yes

2. Has the statistical analysis been performed appropriately and rigorously? 

Reviewer #1: Yes

Reviewer #2: Yes

3. Have the authors made all data underlying the findings in their manuscript fully available?

Reviewer #1: No

Reviewer #2: Yes

4. Is the manuscript presented in an intelligible fashion and written in standard English?

Reviewer #1: Yes

Reviewer #2: Yes

5. Review Comments to the Author

Reviewer #1: I have read this contribution on adding tears to animal faces with great interest! Furthermore, I am convinced that it has merit in demonstrating how tears on animal faces are subject to anthropomorphism mechanisms, and that animal tears may influence perceived aggressiveness and friendliness of animals (even if these relationships are not strong). The paper is generally well written, although I spotted several smaller language issues along the way - i.e., the manuscript could still benefit from some thorough proof-reading.

However, while I think there is reason to be excited about this work, I also saw some issues that limit the extent to which the findings can be interpreted. Perhaps most importantly, there was only a single exemplar for each species, and there was no pre-test data reported for the stimulus selection. E.g., it was unclear if all the baseline animal faces were indeed neutral in expression. Likewise, we do not know how typical these examples were. Would the same results be obtained for slightly different expressions, or different fur colors (affecting tear-visibility) among the same species? Conceptually, I felt that the role of "emotional tears" in this context was not so clear. Were these emotional tears at all? The answer seems to be that this was in the eye of the beholder - however, this would need to be introduced and discussed more clearly. Likewise, the reasoning about the difference between adding tears to neutral animal faces vs. neutral human faces (as in prior work) did not become clear to me. I believe there is a point here, but it would need to be better explained.

Methodologically, I was struggling with the decision to obtain forced choice judgments for the discrete emotions. I find it particularly difficult to interpret any kinds of differences between the baseline images (i.e., the images without the tears that were not digitally manipulated) given this question format. In addition, I believe this should be regarded as a mixed design (with repeated measures across a series of exemplars), even if the exemplars themselves were between-subjects. A related concern here is that the Google-based image selection may have turned up examples that already differed with respect to expressions before the modification. I would therefore find it important to include a breakdown of the results per animal species and emotions (instead of or in addition to figure 3). Finally, relative tear-visibility, and tear-realism may have been a concern here, as the animals differed in both fur color, and the number of tears (one or two). This might be addressed by collecting some post test data for typicality (of the animal pictures), tear-visibility, and tear-realism.

Despite these limitations, I overall found this paper to be very interesting and thought-provoking. It does lend support to the notion that humans might react to animal tears as if they were emotional, and that this might include the type of appeasement functions hypothesized by Hasson and others. I hope that the authors may find my more detailed comments below to be of aid in improving their manuscript.

Introduction:

One of the key arguments for conducting this study is the argument that, since animals do not cry in response to emotional feelings, adding visible tears to neutral emotional expressions presents an excellent opportunity to study the effect of tears. I would tend to agree with this general reasoning. However, the claim that because of this, emotional reactions in observers "can easily be attributed solely to the presence of tears" (p.4) may need some refinement and qualification. It would appear to be a bit more complicated than that. How does the fact that humans are the only species that sheds emotional tears influence the responses of participants? Does this require any conscious reflection by participants? E.g., Media Equation theory would seem to suggest that observers should respond to crying animal stimuli in a similar manner *as if* the tears were produced by humans.

Issues to be considered here are (1) Participants may be more likely to be aware that the tears were digitally added, even provided that the tears look very realistic. It may still be that participants did not question or otherwise consciously reflect upon the presence of the tears in these faces - but if they did, then their emotional responses may also have been affected by this realization. (2) What precisely is the key difference here between seeing tears added to a neutral human face, and seeing tears added to the face of an animal? Animal faces can still be perceived as sad, aggressive etc. (even without the addition of tears). To what extent, and in which way precisely, does the manipulation of the animal faces go beyond adding tears to neutral human faces? I believe the authors have an interesting point here - but I think this needs to be further elaborated and clarified. That said, I agree that it is very interesting to study tears on animals for which we have very different stereotypes (you said "constructs" here - what precisely is meant with that in this context?). Also, the point about daily experience and closer interactions with animals helps to differentiate this from work in HRI/HCI.

When raising the argument about anthropomorphism, I would suggest referring to some of the extant literature on zoomorphism as well. This is not an entirely new field, and it would appear to be highly relevant here. As some of the current references additionally relate to human-robot interaction, this could also include zoomorphic robots, such as Spot (Boston Dynamics), or Miro (Consequential Robotics).

Perhaps it would also help to discuss that the kinds of animals selected in this study are all capable of shedding tears - they just do not use this as an emotional signal to conspecifics. This signalling function appears to be what is included in the process of anthropomorphization.

Visual Stimuli

The example of the dog in figure 1, in my view, looks convincing. However, I have been wondering about some of the other animals. E.g., for the cat, the white fur may have presented a challenge for obtaining comparable visibility/salience of the tears. For the hamster, I wonder if the size of the tears relative to the eyes and the rest of the animal's face might have posed a challenge. I trust that all these materials looked well in the end. However, it would be great if they could be made available in full via supplementary materials.

Measures

Why was the basic emotion item presented as a forced choice? This would appear to exclude the possibility of mixed emotions. Also, I think this makes it difficult to argue that there was "greater sadness" (see abstract). Instead, would this measure not only yield a frequency with which sadness appeared to be dominant?

Procedure

It seems that this was a mixed design, rather than a (pure) between-subjects design - with repeated measurements from each observer for intensity, friendliness, and aggressiveness. A GLMM may be a more suitable statistical approach for these analysis, as it could be used to control for effects of observer identity.

Results

Emotionality - Odds ratios

Figure 3: This figure seems to be missing a scale on the y-axis. The figure caption helps somewhat - but I was still wondering why the squares have different widths.

Is there an explanation for why the results for the cat went in the other direction? Given that there are different stereotypes about these different animals, as well as substantial differences in the basic morphology of their faces, it seems likely that even the tearless images may have been perceived as showing a certain amount of non-neutral expressions. Was there any evidence for this? Did expressions for the tearless animals vary between animals? If yes, then it might be useful to include a figure that shows the baseline emotions for each tearless animal before pooling all of this data.

Intensity

What does it mean that the dog and the chimpanzee "benefited" from the presence of the tears? Do you mean that they were anthropomophized more (more intense perceived emotions)? For friendliness, this is more straight forward.

The gender effects for the horse condition are interesting and seem to align with the notion that personal relevance of the animal should matter.

Figure 4: This figure is much too small and unreadable in the manuscript. The downloadable version was readable, however.

Correlations between variables:

I unfortunately could not find table 1 anywhere in the manuscript, nor in manuscript central. However, the most relevant results appear to have been reported in the text.

SEM:

I found the effects of tears on aggressiveness examined in this analysis to be very interesting. However, it seems that the direct effect, albeit significant, was rather weak. Likewise, the link to perceived friendliness also did not appear to be very strong. Nevertheless, it would seem worth reporting. Perhaps this might be further substantiated in future work, as it could help to make the point that human observers might transfer even some of the purported appeasement functions of this signal to other species.

Unfortunately, I could not access the supplementary materials on OSF without logging in with my ID to request permission from the authors.

Discussion:

The discussion takes up the point again that emotional tears are a solely human feature that non-human animals are unable to produce. I found the argument about extending the work on appeasement signals to be very interesting, and perhaps more clearly formulated here than in the introduction.

However, I found the distinction between emotional and non-emotional tears to be rather fuzzy here. I.e., on the one hand, the argument is made that non-human animals cannot produce emotional tears. On the other hand, shortly thereafter (p.13), it is argued that domestic animals gained benefits from "the emotional stimulus". Please be clearer on how these arguments go together to avoid confusion about when you are talking about emotional tears, and when you are talking about non-emotional tears. Note that the argumentation in the introduction and stimulus design was that tears were added to neutral faces of animals. Where does the emotional stimulus come from then? If the stimuli looked realistic (which I assume they all did), should these tears then not have been non-emotional tears? I think that this potential source of confusion could be addressed - i.e., it may not be necessary for the animals to shed emotional tears for human participants to perceive them as such. Nevertheless, I think would still need to be made more explicitly. I would see the finding that participants perceived these ostensibly non-emotional tears as noteworthy.

The discussion mentions that some of the tears consisted of one teardrop, and some consisted of two teardrops. Given that there was only one exemplar per species, could this distinction have confounded some of the findings - e.g., concerning apparent differences between species? You might refer to prior work in this context that has looked at tear-intensity to address this issue at least partially (i.e., the amount of tears may not matter much). Still, this appears to be a relevant limitation that would call for further research.

On dogs and cats: While I found the discussion about differences in how we relate to cats versus dogs interesting, I felt concerned here that this might be over-interpreting things, given that there was only a single exemplar - a single image, for each. Again, I think it would be important to include in the results also the emotions that were perceived for the unmodified stimuli. Might this specific dog face have been perceived as sadder to begin with, and the cat face as emotional in some other way? Were they both regarded as completely neutral without the tears? I would suggest expanding a bit more on the basic literature about facial expressions in cats and dogs before raising the point about familiarity. E.g., this might start from the DogFACS and CatFacs (both https://www.animalfacs.com/).

SEM: On p.14, it is claimed that the absence of tears is associated with the perception of anger. Perhaps clarify that this discussion point is no longer about the SEM but about the forced-choice emotion rating task. Again, I was also not entirely convinced of this task. Participants saw (purportedly) neutral animal faces with or without tears. I would therefore expect any sense of "anger" in these faces to be rather subtle, and I iwonder if this apparent anger might have been due to slight biases in the selection of images. Were these images pre-tested? -> If the source animal faces were intended to be neutral, then this result would appear to suggest that they were not (entirely) neutral after all. Furthermore, the manipulation involved the addition of tears. Therefore, how does the absence of this manipulation indicate something about the effect of tears in this case? Here, I would be more convinced if there was a set of pre-tested neutral images and, e.g., a Likert-type or continuous scale for each of the discrete emotion. Then, if the addition of tears leads to a reduction of perceived "anger", this would seem to lend further support to Hasson's hypothesis.

Strengths and limitations:

I understood that this is the first study that systematically manipulated tears on animal images, and I would see this as one of the main contributions of this work. However, as in the introduction, I think it would help to elaborate a bit more about how tears on neutral animal faces might differ from examining tears on human faces.

As concerns the limitations, I would see the limited number of exemplars as one of the crucial points. I agree that an excessive number of stimuli might result in habituation effects. However, perhaps there could be two or three exemplars per animal, possibly with somewhat fewer different animals? It could also help to present some additional pre-test or post-test data showing that the chosen exemplars in this study were perceived as both typical and "neutral" exemplars of their species. This is perhaps even emphasized by the findings discussed about animal species and their colors. Here, it would seem that cats, and dogs, and likely also hamsters and horses could be found that have an entirely different fur color. Would, e.g., a black (or dark brown) cat have elicited a different pattern of responses?

Overall, I think that this paper makes a relevant contribution to the literature by showing that humans respond systematically to tears added to animal faces. The majority of the findings appears to show a rather consistent pattern to this effect, and this is strengthened even further by the fact that these animal faces were so heterogeneous overall. However, I would be more cautious when interpreting apparent differences between the species, or the apparent gender effects. The explanations raised for these effects appear plausible, but the present state of the findings seems rather tentative in this regard (mostly because of the very limited number of exemplars).

Future Directions and conclusions

I fully agree with the authors here, that this line of work opens up a lot of interesting possibilities for further research. Likewise, the conclusions very nicely summarize the main points and merits of this work.

Reviewer #2: In this manuscript, the authors attempt to extend the theoretical and empirical literature on emotional tearing in humans to non-human animals (cat, dog, horse, chimpanzee, and hamster). Participant’s gave self-reported ratings of regarding perceptions of emotional intensity as well as individual variables that might affect these perceptions. Similar to the work on tearing in humans, it was found that the presence of tears affected perceptions of in non-humans. Here are my comments/concerns:

1. Lines 109-111 “… however, by blurring vision, tears are a handicap in the case of having to attack or defend against an attacker, as they blur vision.” I think it’s only necessary to mention the blurring of vision once here.

2. The authors mention that emotional tears are uniquely human. I’d like more of a discussion on why there would be expected to be a “tear effect” in non-humans in light of this fact. Of course, the data suggest that there is a reason, but I’d like to see more of a discussion on this.

3. The authors begin the Participants section with a number. I believe this should be written out if they are writing in APA style.

4. Do we have any reason to believe that the digitally added tears were seen as authentic? If not, this could be seen as a limitation.

5. The “F” should be italicized when presenting the F statistic.

6. On line 283, the authors state that “the dog and chimpanzee benefited from the presence of tears…” I think the term “benefited” should be replaced with a less judgmental term. A term that doesn’t imply anything positive or negative. This happens again on line 307.

7. The “r” should be italicized when presenting correlations.

6. PLOS authors have the option to publish the peer review history of their article (what does this mean?). If published, this will include your full peer review and any attached files.

Reviewer #1: No

Reviewer #2: No

---

## [Author Response · Author response to Decision Letter 0]

24 Feb 2021

JOURNAL REQUIREMENTS

Answer: The style requirements are fulfilled 

2. Thank you for including your ethics statement: "The study was under the standards of the Human Research Ethics Committee of the University of Valencia (Spain) and a written informed consent was obtained from all the participants.". 

Please amend your current ethics statement to confirm that your named institutional review board or ethics committee specifically approved this study.

Answer: The statement is amended

Answer: This is done too

Answer: Please consult the cover letter

4. Please include a copy of Table 1 which you refer to in your text on page 12.

Answer: This is done too.

RESPONSES TO REVIEWERS

REVIEWER 1.

We will respond to your comments point by point. First of all, we would like to thank you for your review. We now believe that our work is better because of your review.

Reviewer #1: I have read this contribution on adding tears to animal faces with great interest! Furthermore, I am convinced that it has merit in demonstrating how tears on animal faces are subject to anthropomorphism mechanisms, and that animal tears may influence perceived aggressiveness and friendliness of animals (even if these relationships are not strong). The paper is generally well written, although I spotted several smaller language issues along the way - i.e., the manuscript could still benefit from some thorough proof-reading.

Answer: Thanks very much for your interest in our manuscript and for your encouraging words. We have made deep changes through the whole of the manuscript, taking into account the most of your clever suggestions. Moreover, the new manuscript is now better supported by a new study and its analyses, designed to answer several points from your review. We hope to have clarified also our theoretical concepts, through some new references and background. Please, note that there are many changes in the manuscript (highlighted text). We took note of your suggestion regarding the language, and an expert in scientific English has rechecked our work in order to make it more accurate.

Reviewer #1: However, while I think there is reason to be excited about this work, I also saw some issues that limit the extent to which the findings can be interpreted. Perhaps most importantly, there was only a single exemplar for each species, and there was no pre-test data reported for the stimulus selection. E.g., it was unclear if all the baseline animal faces were indeed neutral in expression. Likewise, we do not know how typical these examples were. Would the same results be obtained for slightly different expressions, or different fur colors (affecting tear-visibility) among the same species? 

Answer: The stimuli we used have been submitted to an additional new study to answer the questions you raised. Please check supplementary materials and new arguments in the main text of the manuscript (including the discussion). Anyway, a legitim doubt (a wondering indeed) is your last question (“Would the same results be obtained for slightly different expressions, or different fur colors (affecting tear-visibility) among the same species?”) which still remains and which has been commented in the limitations of the study.

Reviewer #1: Conceptually, I felt that the role of "emotional tears" in this context was not so clear. Were these emotional tears at all? The answer seems to be that this was in the eye of the beholder - however, this would need to be introduced and discussed more clearly. 

Answer: We believe that this point, which connects clearly with the anthropomorphization conceptual line in the manuscript, has been introduced and discussed much better (please check the Introduction and Discussion sections).

Reviewer #1: Likewise, the reasoning about the difference between adding tears to neutral animal faces vs. neutral human faces (as in prior work) did not become clear to me. I believe there is a point here, but it would need to be better explained.

Answer: Again, we feel that this explanation is improved now.

Reviewer #1: Methodologically, I was struggling with the decision to obtain forced choice judgments for the discrete emotions. I find it particularly difficult to interpret any kinds of differences between the baseline images (i.e., the images without the tears that were not digitally manipulated) given this question format. 

Answer: We understand that perhaps items in Likert scale format could have provided more information, but our choice of an item with discrete emotions was due to our previous work. As we wanted to extend the conclusions of that work to animal faces, it was better not to change the methodology. On the other hand, we do think that it would be necessary to investigate emotions from a dimensional approach, and this could be a great advance that we will consider in our next research.

Reviewer #1: In addition, I believe this should be regarded as a mixed design (with repeated measures across a series of exemplars), even if the exemplars themselves were between-subjects. A related concern here is that the Google-based image selection may have turned up examples that already differed with respect to expressions before the modification. I would therefore find it important to include a breakdown of the results per animal species and emotions (instead of or in addition to figure 3). Finally, relative tear-visibility, and tear-realism may have been a concern here, as the animals differed in both fur color, and the number of tears (one or two). This might be addressed by collecting some post test data for typicality (of the animal pictures), tear-visibility, and tear-realism. 

Answer: We are happy to answer your questions about methodology below. Regarding the choice of the figure, since we had reported the odds ratios by species, we believed that it should provide new information that did not appear in the text in an explicit way. However, those plots can be found in the supplementary material along with the other information.

Reviewer #1: Despite these limitations, I overall found this paper to be very interesting and thought-provoking. It does lend support to the notion that humans might react to animal tears as if they were emotional, and that this might include the type of appeasement functions hypothesized by Hasson and others. I hope that the authors may find my more detailed comments below to be of aid in improving their manuscript.

Answer: We can assure you that your review has been very useful to us, and we believe that thanks to it our work is now better.

Introduction:

Reviewer #1: One of the key arguments for conducting this study is the argument that, since animals do not cry in response to emotional feelings, adding visible tears to neutral emotional expressions presents an excellent opportunity to study the effect of tears. I would tend to agree with this general reasoning. However, the claim that because of this, emotional reactions in observers "can easily be attributed solely to the presence of tears" (p.4) may need some refinement and qualification. It would appear to be a bit more complicated than that. How does the fact that humans are the only species that sheds emotional tears influence the responses of participants? Does this require any conscious reflection by participants? E.g., Media Equation theory would seem to suggest that observers should respond to crying animal stimuli in a similar manner *as if* the tears were produced by humans. 

Answer: We are very grateful for pointing out clues towards a deepening of the scientific literature on anthropomorphization. We have revised and added several references to be considered for the Introduction (some of them regarding Media Equation Theory), so that our argument is now more precise and better supported. Please check the new Introduction section, especially pages 5-6.

Reviewer #1: Issues to be considered here are (1) Participants may be more likely to be aware that the tears were digitally added, even provided that the tears look very realistic. It may still be that participants did not question or otherwise consciously reflect upon the presence of the tears in these faces - but if they did, then their emotional responses may also have been affected by this realization. (2) What precisely is the key difference here between seeing tears added to a neutral human face, and seeing tears added to the face of an animal? Animal faces can still be perceived as sad, aggressive etc. (even without the addition of tears). To what extent, and in which way precisely, does the manipulation of the animal faces go beyond adding tears to neutral human faces? I believe the authors have an interesting point here - but I think this needs to be further elaborated and clarified. 

Answer: We elaborated an argument in which we propose that the presence of the tears (provided they were perceived as realistic from the new study, in supplementary materials) significantly increased the probability for human observers to engage in an anthropomorphization process towards animals, attributing them an emotional expression they did not really have (please check the Discussion section). The extent of whether this process was conscious or unconscious in our observers has not been discussed, mainly because direct measures of anthropomorphism (i.g. questionnaires like ours) can reflect either the implicit or explicit anthropomorphism, or a mixture of both, depending of the person [Złotowski J, Sumioka H, Eyssel F, Nishio S, Bartneck C, Ishiguro H. Model of Dual Anthropomorphism: The Relationship Between the Media Equation Effect and Implicit Anthropomorphism. International Journal of Social Robotics. 2018;10(5):701-714.]. In addition, to deepen in such aspect it was a bit far from the main objectives of our study. 

Reviewer #1: That said, I agree that it is very interesting to study tears on animals for which we have very different stereotypes (you said "constructs" here - what precisely is meant with that in this context?). 

Answer: This phrase was in fact very confusing and it has been reformulated.

Reviewer #1: Also, the point about daily experience and closer interactions with animals helps to differentiate this from work in HRI/HCI.

When raising the argument about anthropomorphism, I would suggest referring to some of the extant literature on zoomorphism as well. This is not an entirely new field, and it would appear to be highly relevant here. As some of the current references additionally relate to human-robot interaction, this could also include zoomorphic robots, such as Spot (Boston Dynamics), or Miro (Consequential Robotics).

Answer: Please check the new references and the mention to such dog-shaped robots in the Discussion section, and we would like to thank you for bringing this to our attention.

Reviewer #1: Perhaps it would also help to discuss that the kinds of animals selected in this study are all capable of shedding tears - they just do not use this as an emotional signal to conspecifics. This signalling function appears to be what is included in the process of anthropomorphization.

Answer: This key point has been added to the Introduction section, together with the reference of Frey’s (1985) book «The mystery of tears»

Visual Stimuli

Reviewer #1: The example of the dog in figure 1, in my view, looks convincing. However, I have been wondering about some of the other animals. E.g., for the cat, the white fur may have presented a challenge for obtaining comparable visibility/salience of the tears. For the hamster, I wonder if the size of the tears relative to the eyes and the rest of the animal's face might have posed a challenge. I trust that all these materials looked well in the end. However, it would be great if they could be made available in full via supplementary materials.

Answer: As you can see in the new version, all materials are available as supplementary material. Following your suggestion of a post-hoc study on the stimuli used, as far as tear visibility is concerned, you can see from the results provided that it was extremely high in all species. 

Measures

Reviewer #1: Why was the basic emotion item presented as a forced choice? This would appear to exclude the possibility of mixed emotions. Also, I think this makes it difficult to argue that there was "greater sadness" (see abstract). Instead, would this measure not only yield a frequency with which sadness appeared to be dominant?

Answer: After reviewing that part of the paper we can only say that we completely agree with you. Our conclusions regarding emotions should be expressed in terms of frequency. We have changed the text to reflect this. Regarding why we present basic emotions as a forced choice, this is because this paper is a continuation of our line of research in which in our previous papers we have included that item in exactly the same way.

Perhaps we should consider opting for a different methodology that allows more flexibility of response, and we intend to be guided by your suggestions for future studies.

Procedure

Reviewer #1: It seems that this was a mixed design, rather than a (pure) between-subjects design - with repeated measurements from each observer for intensity, friendliness, and aggressiveness. A GLMM may be a more suitable statistical approach for these analysis, as it could be used to control for effects of observer identity.

Answer: We have carefully read this comment. As supplementary material we have provided the results of the same study using a fixed and random effects analysis. As can be seen from these results, the variability among participants is minimal, and in essence, the conclusions drawn about the effect of tears on changing perceptions are identical. We opted at first for a simpler analysis because the methodology was the same as that of one of our previous published studies (where the data were analyzed using an ANOVA like the present one), and on the other hand, because we followed the principle of parsimony.

If we were to interpret intensity, friendliness, and aggressiveness as repeated measures, the scores obtained across species should be considered experimental pseudoreplicates (see Lawson, 2015). What we did in this other approach was to introduce species as another random factor along with participants. We could also have opted for a hierarchical model (according to our statistician, an appropriate model) in which the random effects would have been nested. It would not change the conclusions and we would lose ease of interpretability.

Reference

Lawson, J., 2015. Design and Analysis of Experiments with R. 1st ed. Chapman and Hall/CRC.

Results

Emotionality - Odds ratios

Reviewer #1: Figure 3: This figure seems to be missing a scale on the y-axis. The figure caption helps somewhat - but I was still wondering why the squares have different widths.

Answer: In the current version of the manuscript we have added an explanation of how this figure is constructed. Now our readers will be able to know how to obtain the height and width of the bars.

Reviewer #1: Is there an explanation for why the results for the cat went in the other direction? Given that there are different stereotypes about these different animals, as well as substantial differences in the basic morphology of their faces, it seems likely that even the tearless images may have been perceived as showing a certain amount of non-neutral expressions. Was there any evidence for this? Did expressions for the tearless animals vary between animals? If yes, then it might be useful to include a figure that shows the baseline emotions for each tearless animal before pooling all of this data.

Answer: As indicated above, these plots can be found as supplementary material.

Intensity

Reviewer #1: What does it mean that the dog and the chimpanzee "benefited" from the presence of the tears? Do you mean that they were anthropomophized more (more intense perceived emotions)? For friendliness, this is more straight forward.

The gender effects for the horse condition are interesting and seem to align with the notion that personal relevance of the animal should matter.

Figure 4: This figure is much too small and unreadable in the manuscript. The downloadable version was readable, however.

Answer: We agree with you and have changed the graph. We hope that this time the size is right and the colors allow you to see the differences in more detail. In addition, and as we have commented below, the word “benefited” has been removed since it was confusing.

Correlations between variables:

Reviewer #1: I unfortunately could not find table 1 anywhere in the manuscript, nor in manuscript central. However, the most relevant results appear to have been reported in the text.

Answer: In the previous version we made the mistake of not including table 1. This error has now been corrected and Table 1 can be found inserted in the manuscript.

SEM:

Reviewer #1: I found the effects of tears on aggressiveness examined in this analysis to be very interesting. However, it seems that the direct effect, albeit significant, was rather weak. Likewise, the link to perceived friendliness also did not appear to be very strong. Nevertheless, it would seem worth reporting. Perhaps this might be further substantiated in future work, as it could help to make the point that human observers might transfer even some of the purported appeasement functions of this signal to other species.

Unfortunately, I could not access the supplementary materials on OSF without logging in with my ID to request permission from the authors.

Answer: We have fixed that problem and supplementary material can now be accessed anonymously.

Discussion:

Reviewer #1: The discussion takes up the point again that emotional tears are a solely human feature that non-human animals are unable to produce. I found the argument about extending the work on appeasement signals to be very interesting, and perhaps more clearly formulated here than in the introduction.

Answer: We hope that such argument is now clearer in the Introduction and also the Discussion sections.

Reviewer #1: However, I found the distinction between emotional and non-emotional tears to be rather fuzzy here. I.e., on the one hand, the argument is made that non-human animals cannot produce emotional tears. On the other hand, shortly thereafter (p.13), it is argued that domestic animals gained benefits from "the emotional stimulus". Please be clearer on how these arguments go together to avoid confusion about when you are talking about emotional tears, and when you are talking about non-emotional tears. Note that the argumentation in the introduction and stimulus design was that tears were added to neutral faces of animals. Where does the emotional stimulus come from then? If the stimuli looked realistic (which I assume they all did), should these tears then not have been non-emotional tears? I think that this potential source of confusion could be addressed - i.e., it may not be necessary for the animals to shed emotional tears for human participants to perceive them as such. Nevertheless, I think would still need to be made more explicitly. I would see the finding that participants perceived these ostensibly non-emotional tears as noteworthy.

Answer: We noted that the word “benefited” created confusion to the main argument through the text, so we eliminated it. Please, check also that we incorporated the following phrase (from you) to the text in the Discussion section: “it may not be necessary for the animals to shed emotional tears for human participants to perceive them as such”. We think such statement can work as a key point for the final argumentation.

Reviewer #1: The discussion mentions that some of the tears consisted of one teardrop, and some consisted of two teardrops. Given that there was only one exemplar per species, could this distinction have confounded some of the findings - e.g., concerning apparent differences between species? You might refer to prior work in this context that has looked at tear-intensity to address this issue at least partially (i.e., the amount of tears may not matter much). Still, this appears to be a relevant limitation that would call for further research.

Answer: This was just a linguistic figuration to give emphasis to the text, but created confusion, so it has been eliminated from the text. Please check the supplementary materials and see that the tearful images barely differ in the digital manipulation performed (in all animals the tearing was added to both eyes).

Reviewer #1: On dogs and cats: While I found the discussion about differences in how we relate to cats versus dogs interesting, I felt concerned here that this might be over-interpreting things, given that there was only a single exemplar - a single image, for each. Again, I think it would be important to include in the results also the emotions that were perceived for the unmodified stimuli. Might this specific dog face have been perceived as sadder to begin with, and the cat face as emotional in some other way? Were they both regarded as completely neutral without the tears? I would suggest expanding a bit more on the basic literature about facial expressions in cats and dogs before raising the point about familiarity. E.g., this might start from the DogFACS and CatFacs (both https://www.animalfacs.com/).

Answer: Our argument about domestic animals has changed a little in order to be clearer, and, on the other hand, we have indicated that it has to be understood as tentative (thus, as a good line for future research). About the concern on the number of images used to illustrate species, it is commented in the limitations (“However, a possible limitation of the present work is the small number of images used; a greater number of images of each species could have avoided the potential influence of the particularities of each photograph and species.”), and, finally, we didn’t deepen in DogFACS and CatFacs because, though it could be an inspiration for future studies, it was also a bit further from our main objectives in this work.

Reviewer #1: SEM: On p.14, it is claimed that the absence of tears is associated with the perception of anger. Perhaps clarify that this discussion point is no longer about the SEM but about the forced-choice emotion rating task. Again, I was also not entirely convinced of this task. Participants saw (purportedly) neutral animal faces with or without tears. I would therefore expect any sense of "anger" in these faces to be rather subtle, and I iwonder if this apparent anger might have been due to slight biases in the selection of images. Were these images pre-tested? -> If the source animal faces were intended to be neutral, then this result would appear to suggest that they were not (entirely) neutral after all. Furthermore, the manipulation involved the addition of tears. Therefore, how does the absence of this manipulation indicate something about the effect of tears in this case? Here, I would be more convinced if there was a set of pre-tested neutral images and, e.g., a Likert-type or continuous scale for each of the discrete emotion. Then, if the addition of tears leads to a reduction of perceived "anger", this would seem to lend further support to Hasson's hypothesis.

Answer: All the paragraph this comment refers to has been changed and/or deleted in the new Discussion section. Please check also the supplementary study on the quality of the images used.

Strengths and limitations:

Reviewer #1: I understood that this is the first study that systematically manipulated tears on animal images, and I would see this as one of the main contributions of this work. However, as in the introduction, I think it would help to elaborate a bit more about how tears on neutral animal faces might differ from examining tears on human faces.

Answer: The new manuscript takes deepen this last aspect or the study. 

Reviewer #1: As concerns the limitations, I would see the limited number of exemplars as one of the crucial points. I agree that an excessive number of stimuli might result in habituation effects. However, perhaps there could be two or three exemplars per animal, possibly with somewhat fewer different animals? It could also help to present some additional pre-test or post-test data showing that the chosen exemplars in this study were perceived as both typical and "neutral" exemplars of their species. This is perhaps even emphasized by the findings discussed about animal species and their colors. Here, it would seem that cats, and dogs, and likely also hamsters and horses could be found that have an entirely different fur color. Would, e.g., a black (or dark brown) cat have elicited a different pattern of responses?

Answer: This concern has been analysed through the new study on the quality of the images, which we invite you please to consult. However, note that, as we answered above, a legitimate question would still remain about intra-species diversity (in color, fur etc), and such limitation is noted in the manuscript. In this sense, adding more intra-species images would be interesting, but then why not more species? (and so on) … So, we think, taking into account that our main objective goes around the tear effect, the present materials are appropriate enough to reach our goals. 

Reviewer #1: Overall, I think that this paper makes a relevant contribution to the literature by showing that humans respond systematically to tears added to animal faces. The majority of the findings appears to show a rather consistent pattern to this effect, and this is strengthened even further by the fact that these animal faces were so heterogeneous overall. 

Answer: Thanks very much and note that, again, one of your statements has been added to the text (“is strengthened even further by the fact that these animal faces were so heterogeneous overall”).

Reviewer #1: However, I would be more cautious when interpreting apparent differences between the species, or the apparent gender effects. The explanations raised for these effects appear plausible, but the present state of the findings seems rather tentative in this regard (mostly because of the very limited number of exemplars).

Answer: We totally agree with you and both suggestions have been lowered in intensity and stated as tentative.

Future Directions and conclusions

Reviewer #1: I fully agree with the authors here, that this line of work opens up a lot of interesting possibilities for further research. Likewise, the conclusions very nicely summarize the main points and merits of this work.

Answer: Thanks very much for your kindly words and for your time.

REVIEWER 2

We would like to thank you for all your comments. We will answer to your suggestions point by point.

Reviewer #2: In this manuscript, the authors attempt to extend the theoretical and empirical literature on emotional tearing in humans to non-human animals (cat, dog, horse, chimpanzee, and hamster). Participant’s gave self-reported ratings of regarding perceptions of emotional intensity as well as individual variables that might affect these perceptions. Similar to the work on tearing in humans, it was found that the presence of tears affected perceptions of in non-humans. Here are my comments/concerns:

1. Lines 109-111 “… however, by blurring vision, tears are a handicap in the case of having to attack or defend against an attacker, as they blur vision.” I think it’s only necessary to mention the blurring of vision once here.

Answer: We fully agree with you. The sentence has been rewritten so that it is not redundant and the text is clearer.

Reviewer #2:

2. The authors mention that emotional tears are uniquely human. I’d like more of a discussion on why there would be expected to be a “tear effect” in non-humans in light of this fact. Of course, the data suggest that there is a reason, but I’d like to see more of a discussion on this.

Answer: This suggestion has been of great help to us. We have rewritten the discussion so that it is clearer and is an extension of our new introduction. We hope that the reader will understand our explanations in which we focus our hypotheses on the process of anthropomorphization. This topic, along with how visible tears favor emotional inference, has been discussed in greater depth.

Reviewer #2:

3. The authors begin the Participants section with a number. I believe this should be written out if they are writing in APA style.

Answer : We agree with you and have corrected that part of the text to conform to Vancouver standards. 

Reviewer #2:

4. Do we have any reason to believe that the digitally added tears were seen as authentic? If not, this could be seen as a limitation.

Answer: After reading this commentary and some of the comments made by reviewer 1, we found it necessary to conduct a complementary study on the stimuli used. Those results can be found as supplementary material and demonstrate that in general tears were seen as realistic stimuli.

Reviewer #2:

5. The “F” should be italicized when presenting the F statistic.

Answer: We have corrected that part of our results and thank you for your comment.

Reviewer #2:

6. On line 283, the authors state that “the dog and chimpanzee benefited from the presence of tears…” I think the term “benefited” should be replaced with a less judgmental term. A term that doesn’t imply anything positive or negative. This happens again on line 307.

Answer: Thank you very much for this comment. We have realized that we had abused the use of this word. Upon further reading, we have rewritten part of the discussion and replaced the term "benefited" with more appropriate ones.

Reviewer #2:

7. The “r” should be italicized when presenting correlations.

Answer: We have corrected that part of our results and thank you for your comment.

---

## [Decision Letter · Decision Letter 1]

16 Mar 2021

Pécs, Hungary

March 15, 2021

PONE-D-20-35135R1

When Animals Cry: The Effect of Adding Tears to Animal Expressions on Human Judgment

PLOS ONE

Dear Dr. Gadea,

Thank you for submitting your manuscript to PLOS ONE. After careful consideration, we feel that it has merit but does not fully meet PLOS ONE’s publication criteria as it currently stands. Therefore, we invite you to submit a revised version of the manuscript that addresses the points raised by the Reviewer, listed below.

We look forward to receiving your revised manuscript.

Kind regards,

Joseph Najbauer, Ph.D.

Academic Editor

PLOS ONE

Journal Requirements:

Reviewers' comments:

Reviewer's Responses to Questions

**Comments to the Author**

1. If the authors have adequately addressed your comments raised in a previous round of review and you feel that this manuscript is now acceptable for publication, you may indicate that here to bypass the “Comments to the Author” section, enter your conflict of interest statement in the “Confidential to Editor” section, and submit your "Accept" recommendation.

Reviewer #1: (No Response)

2. Is the manuscript technically sound, and do the data support the conclusions?

Reviewer #1: Yes

3. Has the statistical analysis been performed appropriately and rigorously? 

Reviewer #1: Yes

4. Have the authors made all data underlying the findings in their manuscript fully available?

Reviewer #1: Yes

5. Is the manuscript presented in an intelligible fashion and written in standard English?

Reviewer #1: Yes

6. Review Comments to the Author

Reviewer #1: Overall, I would like to thank the authors for having addressed my previous points so thoroughly! Taking this into account, I find the present version of this manuscript to be substantially improved. In particular, I appreciated the inclusion of the additional post-test and analyses included in the supplementary materials that, I think, are very helpful with respect to better understanding the strengths and limitations of the present work. Likewise, I would like to thank the authors for having performed additional fixed and random effects analyses. The finding that these results converged with the previous ANOVA-results suggest that these were indeed robust. It also makes sense to me to stick with presenting the simpler ANOVA results in this case.

As illustrated by the post-test, a few limitations still remain: the limited number of exemplars; the lack of realism for a few of the stimuli (horse, cat); the variability in the extent to which some of the tearless stimuli were perceived as emotional (e.g., horse vs. dog). I therefore still find some parts of the discussion regarding differences between animal species (e.g., between cats and dogs) to perhaps be a bit premature at this point. Nevertheless, it is of course interesting to speculate about such differences, and this is being done more cautiously now (and in view of the limitations of this initial study). The discussion has furthermore been substantially improved overall, and the main findings are presented more clearly now.

Results of the new rating study:

I believe that the new post-test has helped to sufficiently strengthen this already very interesting paper. Since these materials are openly available, it should be easy to replicate/extend the current work. Tears were overall judged as very well visible in the new rating study. This, together with being able to view all the pictures in full helps to address a lot of the concerns about the technical soundness of this work. However, as shown by the study of the stimuli, some limitations do remain. First, perceived emotionality of the tearless images appears to have been quite variable. In particular, the tearless horse picture seems to have been perceived as substantially less emotional than the dog picture (which was already perceived as very emotional even without tears). This suggests that some of the effects of tears on specific animal images may indeed be more difficult, and perhaps more interesting, to interpret (see discussion). Second, it seems that the tears on the Horse and Cat may not have been perceived as sufficiently realistic. I.e., here, it would seem that the answer to this question from the response letter: "provided they were perceived as realistic from the new study" - may have to be that this is not fully supported by the post-test. Nevertheless, neither of these limitations would seem to invalidate/threaten the conclusion that the addition of tears generally increased perceived emotionality.

I only have a few minor additional comments beyond the above-mentioned limitations of the post-test:

Introduction:

I was not sure if quite as much background on human eye gaze is needed here. The first paragraph might be shortened somewhat to move more quickly towards tears, or animal crying.

Regarding reference [9], i.e., responses to tears within 50 milliseconds - here, recent work has so far failed to replicate this finding (Gračanin et al., 2021).

Reference:

Gračanin, A., Krahmer, E., Balsters, M., Küster, D., & Vingerhoets, A. J. (2021). How weeping influences the perception of facial expressions: The signal value of tears. Journal of Nonverbal Behavior, 45(1), 83-105.

Minor points:

p.5, l.134: there is an opening "(" here that is missing the closing ")".

p.5, l.154: replace "according their lacking of..." with "according to their lack of..."

p.6, l.174: replace "for such process" with "for such [a] process"

p.16, l.450: replace "confirmed our hypothesis" with "supported our hypothesis"

7. PLOS authors have the option to publish the peer review history of their article (what does this mean?). If published, this will include your full peer review and any attached files.

Reviewer #1: No

---

## [Author Response · Author response to Decision Letter 1]

12 Apr 2021

Journal Requirements:

RESPONSE: The reference list is revised. No retracted papers. A reference was changed after reading a comment from the reviewer (please check the response below) 

Comments to the Author

Reviewer #1: Overall, I would like to thank the authors for having addressed my previous points so thoroughly! Taking this into account, I find the present version of this manuscript to be substantially improved. In particular, I appreciated the inclusion of the additional post-test and analyses included in the supplementary materials that, I think, are very helpful with respect to better understanding the strengths and limitations of the present work. Likewise, I would like to thank the authors for having performed additional fixed and random effects analyses. The finding that these results converged with the previous ANOVA-results suggest that these were indeed robust. It also makes sense to me to stick with presenting the simpler ANOVA results in this case.

As illustrated by the post-test, a few limitations still remain: the limited number of exemplars; the lack of realism for a few of the stimuli (horse, cat); the variability in the extent to which some of the tearless stimuli were perceived as emotional (e.g., horse vs. dog). I therefore still find some parts of the discussion regarding differences between animal species (e.g., between cats and dogs) to perhaps be a bit premature at this point. Nevertheless, it is of course interesting to speculate about such differences, and this is being done more cautiously now (and in view of the limitations of this initial study). The discussion has furthermore been substantially improved overall, and the main findings are presented more clearly now.

RESPONSE: Regarding the dog, we are in agreement with you in that it was seen as emotional even in the original tearless image, which we described as «unique » in the discussion, given the amount of emotionality rated on it. At this regard, we think it could be because a variety of reasons, including our relationship with dogs in general (as especulated in the Discussion) or also due to the chosen breed for the photography, or even to the expression of this concrete dog/photo. In any case, this illustrates the need for a careful selection of the images, and also the need of a quality study like the one we performed, following your suggestion. Please check a couple of new phrases included in the dog&cat argument as advices for caution with interpretations, and also a new phrase in the « Strenghts and limitations » section, in line with your comments.

Results of the new rating study:

I believe that the new post-test has helped to sufficiently strengthen this already very interesting paper. Since these materials are openly available, it should be easy to replicate/extend the current work. Tears were overall judged as very well visible in the new rating study. This, together with being able to view all the pictures in full helps to address a lot of the concerns about the technical soundness of this work. However, as shown by the study of the stimuli, some limitations do remain. First, perceived emotionality of the tearless images appears to have been quite variable. In particular, the tearless horse picture seems to have been perceived as substantially less emotional than the dog picture (which was already perceived as very emotional even without tears). This suggests that some of the effects of tears on specific animal images may indeed be more difficult, and perhaps more interesting, to interpret (see discussion). Second, it seems that the tears on the Horse and Cat may not have been perceived as sufficiently realistic. I.e., here, it would seem that the answer to this question from the response letter: "provided they were perceived as realistic from the new study" - may have to be that this is not fully supported by the post-test. Nevertheless, neither of these limitations would seem to invalidate/threaten the conclusion that the addition of tears generally increased perceived emotionality.

RESPONSE: The phrase you cite -copied from the rebuttal letter- is certainly not fully supported by the post-test (surely due to an excess of abstract for the letter), and in fact it was not included in the text of the manuscript. Instead, in the anterior manuscript it can be read: “Finally, we noted a few unexpected results (the “colder tearful cat” and the absence of a tear effect on aggressiveness in the horse) that were obtained with the pictures that were rated as less realistic in the quality study. This prompts an interesting reflection on the use of artificial stimuli to depict human emotions in non-human targets (animals and possibly robots), in the sense that, the more artificial and unreal they are perceived to be, the more difficult it is for them to produce empathy in us, and thus an anthropomorphic feeling” We think that this paragraph is expressing the limitations you note in a constructive manner. 

I only have a few minor additional comments beyond the above-mentioned limitations of the post-test:

Introduction:

I was not sure if quite as much background on human eye gaze is needed here. The first paragraph might be shortened somewhat to move more quickly towards tears, or animal crying.

RESPONSE: The first paragraph has been adequately shortened.

Regarding reference [9], i.e., responses to tears within 50 milliseconds - here, recent work has so far failed to replicate this finding (Gračanin et al., 2021).

Reference:

Gračanin, A., Krahmer, E., Balsters, M., Küster, D., & Vingerhoets, A. J. (2021). How weeping influences the perception of facial expressions: The signal value of tears. Journal of Nonverbal Behavior, 45(1), 83-105.

RESPONSE: We thank you for this reference, which we have read with great interest. After carefully reading, we have decided to change a bit our sentence and to eliminate the commented reference (number 9: Balsters et al., 2013). We noted that both the Balsters paper and the attempt to replicate its findings by Gračanin et al. assess reaction times and not the influence of a preattentional stimulus. Given that our aim was not the measurement of exact reaction times, and that this paragraph of the Introduction was written just to generally point that we (and our brain) in fact react to emotional tears, we now cite a reference we found in the literature on bioelectrical changes caused by the sight of tears: Krivan S, Caltabiano N, Cottrell D, Thomas N. I'll cry instead: Mu suppression responses to tearful facial expressions. Neuropsychologia. 2020;143:107490.

Minor points:

p.5, l.134: there is an opening "(" here that is missing the closing ")".

p.5, l.154: replace "according their lacking of..." with "according to their lack of..."

p.6, l.174: replace "for such process" with "for such [a] process"

p.16, l.450: replace "confirmed our hypothesis" with "supported our hypothesis"

RESPONSE: All these minor points have been observed and corrected.

---

## [Decision Letter · Decision Letter 2]

20 Apr 2021

Pécs, Hungary

April 19, 2021

When Animals Cry: The Effect of Adding Tears to Animal Expressions on Human Judgment

PONE-D-20-35135R2

Dear Dr. Gadea,

We’re pleased to inform you that your manuscript (R2 version) has been judged scientifically suitable for publication and will be formally accepted for publication once it meets all outstanding technical requirements.

Kind regards,

Joseph Najbauer, Ph.D.

Academic Editor

PLOS ONE

Reviewers' comments:

Reviewer's Responses to Questions

**Comments to the Author**

1. If the authors have adequately addressed your comments raised in a previous round of review and you feel that this manuscript is now acceptable for publication, you may indicate that here to bypass the “Comments to the Author” section, enter your conflict of interest statement in the “Confidential to Editor” section, and submit your "Accept" recommendation.

Reviewer #1: All comments have been addressed

2. Is the manuscript technically sound, and do the data support the conclusions?

Reviewer #1: Yes

3. Has the statistical analysis been performed appropriately and rigorously? 

Reviewer #1: Yes

4. Have the authors made all data underlying the findings in their manuscript fully available?

Reviewer #1: Yes

5. Is the manuscript presented in an intelligible fashion and written in standard English?

Reviewer #1: Yes

6. Review Comments to the Author

Reviewer #1: (No Response)

7. PLOS authors have the option to publish the peer review history of their article (what does this mean?). If published, this will include your full peer review and any attached files.

Reviewer #1: **Yes: **Dennis Küster

---

## [Editor Report · Acceptance letter]

26 Apr 2021

PONE-D-20-35135R2 

When animals cry: The effect of adding tears to animal expressions on human judgment 

Dear Dr. Gadea:

I'm pleased to inform you that your manuscript has been deemed suitable for publication in PLOS ONE. Congratulations! Your manuscript is now with our production department. 

Kind regards, 

on behalf of

Dr. Joseph Najbauer 

Academic Editor

PLOS ONE